# Direct and indirect effects of age on dengue severity: The mediating role of secondary infection

Esther Annan[1]*, Jesús Treviño[2], Bingxin Zhao[3], Alfonso J. Rodriguez-Morales[4,5,6], Ubydul Haque[7,8]

**1** Center for Health and Wellbeing, School of Public and International Affairs, Princeton University, Princeton, New Jersey, United States of America, **2** Department of Urban Affairs at the School of Architecture, Universidad Autónoma de Nuevo León, San Nicolás de los Garza, Nuevo Léon, México, **3** Department of Statistics and Data Science, The Wharton School, University of Pennsylvania, Philadelphia, Pennsylvania, United States of America, **4** Faculty of Health Sciences, Universidad Científica del Sur, Lima, Peru, **5** Gilbert and Rose-Marie Chagoury School of Medicine, Lebanese American University, Beirut, Lebanon, **6** Grupo de Investigación Biomedicina, Faculty of Medicine, Fundación Universitaria Autónoma de las Américas-Institución Universitaria Visión de las Américas, Pereira, Risaralda, Colombia, **7** Rutgers Global Health Institute, New Brunswick, New Jersey, United States of America, **8** Department of Biostatistics and Epidemiology, School of Public Health, Rutgers University, Piscataway, New Jersey, United States of America

\* esther.annan@princeton.edu

**Data Availability Statement:** Data can be found at 10.5281/zenodo.8169770 (https://doi.org/10.5281/zenodo.8169770).

## Abstract

Severe dengue occurrence has been attributed to increasing age and different dengue virus (DENV) serotypes that cause secondary infections and immune-enhancing phenomena. Therefore, we examined if the effect of age on dengue severity was mediated by infectivity status while controlling for sex and region. Further, we assessed the spatial clustering of dengue severity for individuals with primary and secondary infection across Mexican municipalities. Health data from 2012 to 2017 was retrieved from Mexico's Ministry of Health. A mediation analysis was performed using multiple logistic regression models based on a directed acyclic graph. The models were explored for the direct effect of age on dengue severity and its indirect impact through secondary infection. In addition, severe dengue clusters were determined in some Northeastern and Southeastern municipalities through spatial analysis. We observed a nonlinear trend between age and severe dengue. There was a downward trend of severe dengue for individuals between 0 and 10 years and an upward trend above 10 years. The effect of age on dengue severity was no longer significant for individuals between 10 and 60 years after introducing infectivity status into the model. The mediating role of infectivity status in the causal model was 17%. Clustering of severe dengue among individuals with primary infection in the Northeastern region may point to the high prevalence of DENV-3 in the region. Public health efforts may prevent secondary infection among infants and the aged. In addition, there should be a further investigation into the effect of DENV-3 in individuals with primary disease.

**Funding:** The author(s) received no specific funding for this work.

## Author summary

Age has been previously found to be associated with severe dengue. However, it is unclear whether this effect is purely due to aging or the predisposition of older individuals to be exposed to secondary infection. Since secondary infection with a dengue serotype other than the infecting serotype during the first infection increases the risk of severe dengue, we explored how much this effect contributed to the effect of age on severe dengue. We performed a mediation analysis and assessed the spatial distribution pattern of primary and secondary infection across Mexico to determine regions vulnerable to severe dengue occurrence. The youngest and oldest population groups had relatively higher odds of severe dengue. However, the effect of age on severe dengue was partly mediated by secondary infection. The spatial analysis also revealed aggregation of severe dengue cases among individuals with primary infection in the Northeastern region, which may be associated with the high prevalence of DENV-3 in the region. Focusing on the prevention of secondary infection among the elderly and infants may help to avert the number of severe dengue cases.

## Introduction

Dengue fever has in recent years expanded in geographic distribution, causing endemic diseases with seasonal fluctuations across the globe [1–4]. In Mexico, the force of dengue infection between 2008 and 2014 was about 7.1% (5.1%–9.8%), indicating a high endemic transmission [5]. Infection with dengue presents on a spectrum of mild to severe, with economic impact primarily attributable to indirect costs from loss of productivity [6]. The seasonal transmission of dengue may be explained by environmental factors [7], viral evolution, and population-specific immunity against different dengue virus (DENV) serotypes [8]. Individual-level predictors of disease severity have been found to include age, comorbidities, sex, and infection with multiple serotypes [9–14].

After a first epidemic, subsequent epidemics result in fewer cases due to herd immunity, and younger individuals become progressively susceptible as the immune population ages [15]. As epidemics continue over time, the age-specific seroprevalence is expected to change, with small outbreaks involving younger generations [15]. The risk of having a classical dengue disease, which traditionally reflects less severe symptoms, has been found to increase with age after a person experiences primary infection [16].

Primary dengue infection is characterized by high titers of immunoglobulin M (IgM) in 3–5 days and immunoglobulin G (IgG) antibodies from the sixth to the tenth day after symptom onset [17]. While the IgM disappears in 2–3 months, IgG persists in the body for life, providing lifelong immunity against the infecting serotype, but not any of other the three dengue serotypes [17]. When a person is re-infected with a different or unencountered DENV serotype, this usually leads to classical dengue fever (DF) or dengue without warning signs [17]. Of the people with a secondary infection, 2% - 3% progress to dengue with warning signs, which may further result in severe dengue and death [17]. DHF is characterized by hemorrhagic signs like thrombocytopenia, petechiae and epistaxis, while DSS occurs due to leakage of intravascular fluids and proteins into perivascular spaces [18]. Antibody-dependent enhancement (ADE) occurs because of secondary infection with a heterologous DF serotype and is associated with more severe infections. However, not all severe infections are a result of a secondary infection [17].

An increase in an individual's age increases the likelihood of exposure to a secondary infection, and the outcome of dengue serotype 2 (DENV-2) secondary infection has been found to

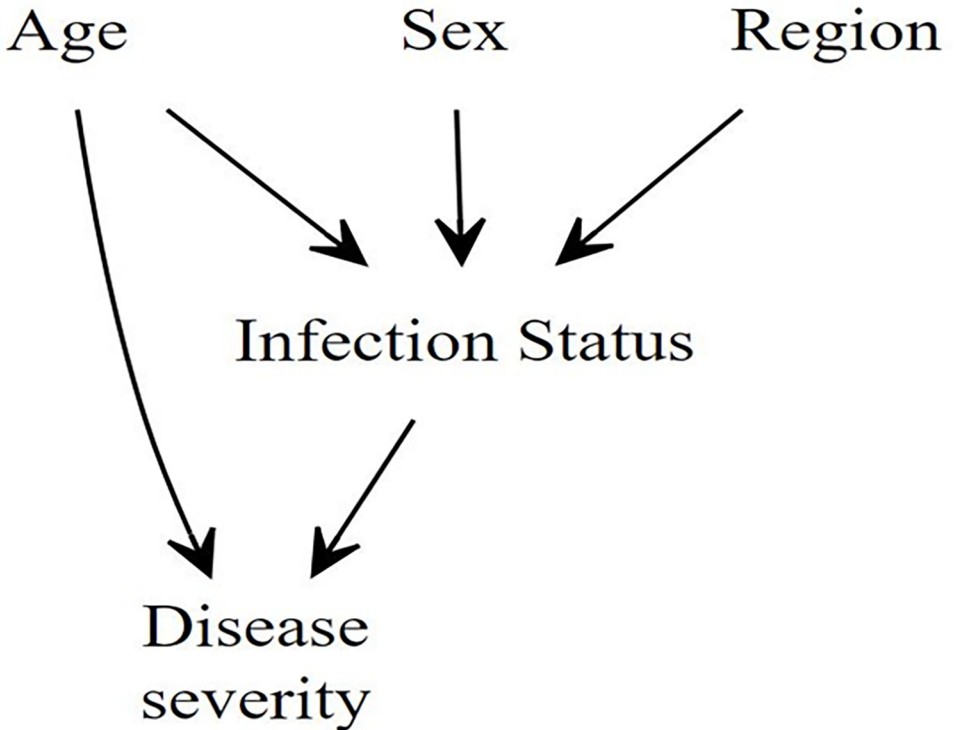

**Fig 1. Directed Acyclic Graph (DAG) of factors associated with dengue severity.**

cause an increase in the risk of severe clinical presentation of DF [19]. The association between secondary serologic response and severe dengue has been established in the previous literature [20], and prior studies have explored the influence of age on clinical dengue presentation. Age has been found to be associated with an increased risk of clinical dengue [1]. However, to the best of our knowledge, the mediation effect of secondary infection in the causal pathway between age and severe dengue has not been studied. Hence, our study aimed to quantify the direct and indirect effect of age on severe dengue through the mediating role of secondary infection (Fig 1). This mediation analysis was performed to help provide information about the causal mechanism between age and dengue severity. We hypothesized that the effect of age on dengue severity would be insignificant after the introduction of infectivity status into a model with which age causes severe dengue, after controlling for confounding variables. We further characterized the spatial distribution of infectivity status for severe dengue across Mexico. While a secondary infection of dengue may be due to a homologous infection, a secondary infection that is associated with severe dengue is likely due to infection with a heterologous DENV serotype [21]. Hence, we expected to find spatial differences across infectivity status for those with severe dengue. Regional differences of infectivity may further be associated with a differential presence of DENV serotypes, which may potentially affect efforts to control dengue in areas with significant clusters of secondary infection.

## Methods

### Ethics statement

This study was reviewed and approved by the ethics and research committee of the Universidad Autonoma de Nuevo Leon and "North Texas Regional Institutional Review Board" as an exempt category (IRB# 2021–035).

## Data collection and preprocessing

Data used for the analysis was acquired from the Ministry of Health, Mexico. Our population of interest included all individuals diagnosed with dengue in Mexico with information about IgM and IgG serology results from 0 to 100 years. Individuals with missing data on IgM, IgG laboratory results and/or dengue diagnosis were excluded from the analysis. The dataset had information about individuals meeting the inclusion criteria from 2012 to 2017. Hence, data for all variables of interest were compiled from 2012 to 2017, and a cross-sectional study was performed. The analysis was restricted to observations from these six years. The Ministry of Health data reports non-identifiable clinical findings from notifiable health units in 2,469 Mexican municipalities. This study received formally written approval as an exempt category by the North Texas Regional Institutional Review Board and the Universidad Autónoma De Nuevo León.

## Definition of variables

**Outcome variable.** Individuals with signs of clinical dengue were further tested for laboratory-confirmed evidence of non-structural protein (NS1) of DENV. An individual with positive NS1 results was classified as having dengue. DF is classified in the dataset as non-severe dengue, severe dengue, and dengue with alarming signs. These classifications are closely synonymous with DF, dengue hemorrhagic fever (DHF), and dengue shock syndrome (DSS), respectively [22]. DF severity was categorized as non-severe dengue and severe dengue. Based on the world health organization's emphasis on critical care for individuals with alarming signs, people with serious dengue or dengue with alarming signs had severe dengue, while those with non-serious dengue were in the 'non-severe dengue' group [23]. Disease classifications other than the DF categories were grouped as 'other' in the dataset. Data entered as 'other' or missing were excluded from the analysis.

**Exposure variable.** Age was used in the analysis as a continuous variable.

**Mediator variable.** A person with IgG positive DF and a current DF diagnosis and IgM positive result were classified as having a secondary infection. Individuals with IgM positive, as well as IgG negative results were classified as having only a primary infection and no secondary infection.

**Covariates.** Demographic variables reported in the dataset include sex and state. Sex was categorized as male or female. States were further categorized into regions. Baja California, Baja California Sur, Sonora, Sinaloa, and Chihuahua were defined as the 'Northwest' region. Durango, Coahuila, Nuevo Leon, and Tamaulipas were defined as 'Northeast' region. Zacatecas, San Luis Potosi, San Luis Potosi, Aguascalientes, Guanajuato, Queretaro, Nayarit, Jalisco, Colima, and Michoacan were categorized as 'Center west' region. Hidalgo, Mexico City, Distrito Federal, Morelos, Puebla, Tlaxcala were in the 'Center' region, while Oaxaca, Guerrero, Veracruz, Chiapas, Tabasco, Campeche, Quintana Roo and Yucatan were classified under 'Southeast' region.

## Statistical analysis

The proportion of dengue severity across demographic variables were determined. Variances across subgroups were tested for statistical significance using chi-square tests for categorical variables and independent sample t-tests for continuous variables. The Type I error alpha value was set at 0.05 for all significance tests. Univariate analyses were conducted for descriptive purposes, and bivariate analyses were performed to determine associations between different independent variables and dengue severity. All bivariate analyses were adjusted for False Discovery Rate (FDR). Since the relationship between age and severe dengue was found to be

non-linear, a generalized additive model (GAM) was used to model the relationship between age and severe dengue. The 'PROC GAMPL' SAS procedure which fits GAMs based on low-rank regression splines [24] was used for modeling this relationship. Missing data included 51386 observations that were excluded from the final sample. The final sample size for the analysis was 52,212.

## Mediation analysis

The mediation analysis comprised the assessment of 1) the direct effect of the age variable on a binary dengue severity variable and 2) the indirect effect of age on a binary dengue severity variable through a binary infectivity status variable. Two traditional generalized linear regression-based approaches have been used for mediation analysis in prior literature: a difference method and a product method [25]. When modeling a binary outcome, the estimates from the difference method vary from the product method due to non-collapsibility of the exposure-outcome effect across mediator values [26,27]. Further, the difference-in-coefficients method conflates the indirect effect when modeling a binary outcome [28]. Hence, the product-of-coefficients was used in our analysis. When using the product method in the mediation analysis, the procedure comprises the estimation of two regression equations; Eq 1 evaluates the effect of the exposure (x) and covariate (c) on the mediator (m), while Eq 2 estimates the effect of x, m and c on the outcome (Y) [25].

$$\text{logit } \{P\,(M = 1|a,\ c)\} = \beta_0 + \beta_1 x + \beta\prime_2 c \tag{1}$$

$$\text{logit } \{P\,(Y = 1|x,\ m,\ c)\} = \theta_0 + \theta_1 x + \theta_2 m + \theta_3 c \tag{2}$$

where, respectively, $\beta_0 + \beta_1$ and $\beta\prime$ represent the intercept and effects of age (x) and confounder (c) on infectivity status. Similarly, $\theta_0$, $\theta_1$, $\theta_2$, and $\theta_3$ represent the intercept and respective effects of age (x), infectivity status (m) and confounder (c) on disease severity classification. The mediation analysis was done using the SAS CAUSALMED procedure. The CAUSALMED procedure employs the counterfactual framework [29], where direct and indirect effects are defined in terms of counterfactual outcomes. The controlled direct effect (CDE) is the difference between the response and non-response outcome for a specific mediator value (3). The natural direct effect (NDE) is the difference between the response outcomes when the effect of the mediator is set to 0 (4), while the natural indirect effect (NIE) is the difference between the mediator levels when the response outcome is present (5) [30].

$$\text{CDE} = Y_1 m - Y_0 m \tag{3}$$

$$\text{NDE} = Y_1 m_0 - Y_0 m_0 \tag{4}$$

$$\text{NIE} = Y_1 m_1 - Y_1 m_0 \tag{5}$$

Eq (6) shows the formula for the total effect (TE) decomposition and the percentage of total effect that is mediated (PM), is therefore calculated using the formula in Eq (7). The definitions of these effects are applicable to both linear and nonlinear models since they are independent of the outcome and mediator [30].

$$\text{TE} = \text{NDE} + \text{NIE} \tag{6}$$

$$\text{PM} = \text{NIE}/\text{TE} * 100\% \tag{7}$$

Based on the directed acyclic graph (DAG) (Fig 1), several models were explored to

determine the direct effect of age on dengue severity, the indirect effect of age mediated by infectivity status on dengue severity and the effect of age on infectivity status. For each model, sex and region were used as covariates. Based on the Akaike Information Criterion (AIC) and likelihood ratio test, interactions were explored, and the model without interactions was found to be the best model for the estimation of the direct and indirect effects.

### Spatial analysis

Severe and non-severe dengue cases were summed up by secondary infection status. That was used in a cluster analysis where the spatial variability of severe dengue was compared by infection status across Mexican municipalities. The spatial distribution of severe and non-severe dengue was visualized for both primary and secondary infection. An optimized outlier analysis of severe dengue was then performed for each infection status. Statistically significant spatial clusters of high values (hot spots) and low values (cold spots) were identified using the Anselin Local Moran's I (LMi Index) statistic. The Anselin local Moran's I is a local indicator of spatial association (LISA). The LISA for an observation shows the extent to which there is significant spatial clustering of similar values around that observation, while the sum of all LISAs correlated with the global spatial association [31]. The statistics behind the Anselin formula is further explained elsewhere [32]. The FDR correction method was used at a 5% error rate to correct for multiple testing and spatial dependence [33].

### Coding and environment

Preprocessing of data, analysis, and generation of figures were done using SAS (version 9.4, SAS Institute Inc., Cary, NC, USA).

### Results

Of the 52,212 individuals with reported data, 60% were female, with roughly half (50%) living in the Southeast region and 81.2% having non-severe dengue (Table 1). The average age was 32 years, ranging from 0 to 100 years, with females being averagely older (33 years) than males (30 years). The difference between primary and secondary infection by gender was not statically significant ($\chi^2$ = 0.007, p = 0.9447 (S1 Table). On average, individuals with secondary infection were older than those with primary infection (S1 Fig). This difference was statistically

**Table 1. Demographic characteristics by dengue classification.**

| Parameter | | Dengue Classification, n (%) | | |
|---|---|---|---|---|
| | | Severe Dengue | Non-Severe Dengue | All |
| **Sex | Male | 4645 (31.42%) | 16243 (77.76%) | 20888 (40.01%) |
| | Female | 5181 (16.54%) | 26143 (83.46%) | 31324 (59.99%) |
| **Region | Center | 479 (4.87%) | 2156 (5.09%) | 2635 (5.05%) |
| | Center- West | 966 (9.83%) | 6855 (16.17%) | 7821 (14.98%) |
| | North—East | 1173 (11.94%) | 11100 (26.19%) | 12273 (23.51%) |
| | North—West | 435 (4.43%) | 2837 (6.69%) | 3272 (6.27%) |
| | South—East | 6773 (68.93%) | 19438 (45.86%) | 26211 (50.20%) |
| NS Age in years (SD) | | 32 (±19.80) | 32 (±18.71) | 32 (±18.92) |
| Total (n) | | 9826 (18.18%) | 42386 (81.18%) | 52212 (100%) |

** Chi-square performed for group differences had p <0.001

NS P-value of T-test not significant

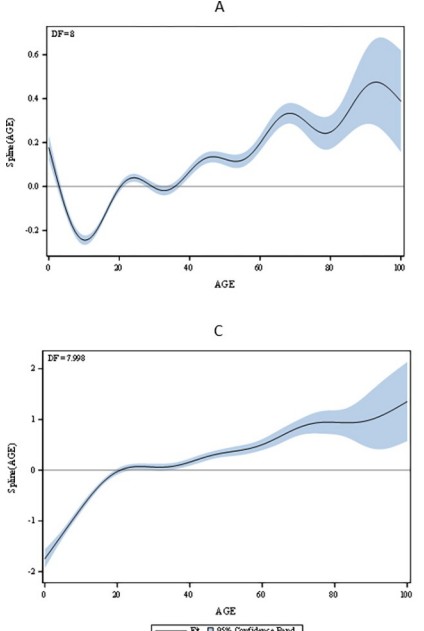
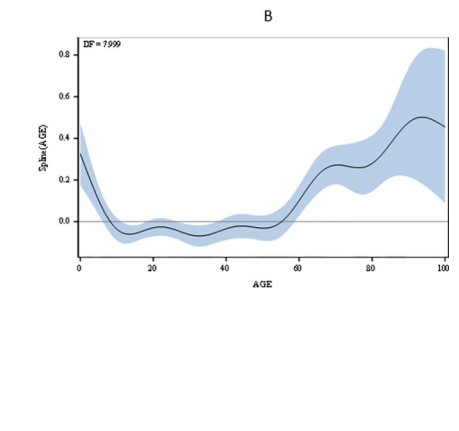

**Fig 2. Regression splines showing a. the direct relationship between age and severe dengue b. the indirect relationship between age and severe dengue, mediated by infectivity status c. the relationship between age and infectivity status.**

significant [t $(_{7961.7})$ = -15.99, p $<$ 0.0001]. The prevalence of secondary infection was higher among individuals with severe dengue (95%) than those with non-severe dengue (87%). Descriptive statistics for the age category are in Tables 1 and S1. A t-test showed that there was no statistically significant difference across age for dengue severity. However, extreme age categories (youngest and oldest) were significantly associated with higher proportions of disease severity (Fig 2A and 2B). On average, individuals with secondary infection were older than individuals with primary infection (t-value = -15.87, p-value <0.0001).

S2 and S3 Figs show the proportions of severe (a) and non-severe (b) dengue and the proportions of infection status for each category. Hence, all four proportions add up to 100% for every age. Compared to the other ages, there was a lower frequency of secondary infection among individuals about 0–5 years for both severe and non-severe dengue (S2A and S2B Fig). The average frequency of dengue among those with secondary infection from about five years to 60 years was about 20% and 70%, respectively, for those with severe and non-severe dengue. However, above 60 years, there was a steeper increase in the frequency among those with severe dengue than those with non-severe dengue (S2A and S2B Fig). The frequency of non-severe dengue decreased with age among those with primary infection. However, the frequency of severe dengue among those with primary infection and severe dengue remained around 5% for the entire age distribution except for those above 60 years who had higher frequencies of severe dengue (S2A and S2B Fig). Differences observed between primary and secondary infection for severe dengue (t-value = -2.56, p = 0.001) was significant but not for non-severe dengue (t-value = -0.99, p = 0.3219). S3A and S3B Fig show variations in the frequency of primary and secondary infection from 2012 to 2017 for severe and non-severe dengue. Cochrane-Armitage trend test for the difference between primary and secondary infection for severe dengue (Z = 2.01, p = 0.04) and non-severe dengue (Z = -14.67, p<0.0001) were both statistically significant. For those with primary infection, there is an observed increase in the proportion of severe and non-severe dengue between 2016 and 2017. Although secondary

**Table 2. Proportions and p-value for different chi-squared test results with 95% significance.**

| Parameter | | Dengue Classification (%) | | |
|---|---|---|---|---|
| | | Severe Dengue | Non-Severe Dengue | Chi-square p-value |
| Serotype | DENV- 1 | 58 (0.59%) | 233 (0.55%) | < .0001 |
| | DENV- 2 | 183 (1.86%) | 289 (0.68%) | |
| | DENV- 3 | 5 (0.05%) | 18 (0.04%) | |
| | DENV- 4 | 3 (0.03%) | 37 (0.09%) | |
| | Missing | 9577 (97.47%) | 41809 (98.64%) | |
| Secondary infection | Yes | 9333 (94.98%) | 36735 (86.67%) | < .0001 |
| | No | 493 (5.02%) | 5651 (13.33%) | |

infection was the commonest in the Southeast region (55.8%), primary infection was commonest in the Northeast region (80.5%) (S1 Table and S4 Fig). Regional differences were statistically significant [$\chi^2$ = 12775.3, p< 0.0001].

DENV-2 was the commonest serotype among individuals with severe and non-severe dengue (Table 2). The serotype distribution pattern was similar for both sexes (S5 Fig). However, the proportion of those with secondary infection who had severe dengue was higher than those with non-severe dengue. Observed serotype and infection status differences were statistically significant (Table 2). The Southeast region had the highest proportion of DENV-2, DENV-3, and DENV-4 and the second highest for DENV-1 (S2 Table). Apart from the Southeast region, the Northeast had the highest proportion of DENV-3 and a higher proportion of DENV-4. In contrast, the Center west region had higher proportions of DENV-1 and DENV-2 than the other regions.

S3–S5 Tables show the results from the GAM procedures. When controlled for other variables, males have 0.91 (CI: 0.86, 0.97) times the odds of females of having a secondary infection (S3 Table). However, compared to females, males tend to have higher odds [1.40 (CI: 1.35, 1.46)] of severe infection after controlling for region and infectivity status (S5 Table). While there exists higher odds of secondary infection in the Southeast [2.19 (CI: 1.76, 2.73)] and Northwest [3.10 (CI:2.30, 4.51)] regions compared to the Center region (S3 Table), the odds of severe dengue is higher for the Center region compared to all regions, except for the Southeast region (S5 Table). Fig 2A and 2B show the GAM with severe diagnosis as the outcome variable. While Fig 2A models age on severe diagnosis, Fig 2B models age and infection status on dengue severity. Addition of infectivity status in model 2b resulted in a general non-significant pattern of association between individuals within the age ranges of 10 and 60 years and severe dengue. Both models, however, show a clear pattern of decreasing severity from 0 to about 10 years, and an increasing pattern above 60 years of age, although the confidence intervals also widened with increasing age. Fig 2C shows a steep increase in the association between age and

**Table 3. Results from mediation analysis.**

| | Estimate | Standard Error | LCI (Lower Confidence Limit) | UCI (Upper Confidence Limit) | P-Value |
|---|---|---|---|---|---|
| Total Effect | 0.0005 | 0.0001 | 0.0003 | 0.0007 | < .0001 |
| Controlled Direct Effect (CDE) | 0.0004 | 0.0001 | 0.0002 | 0.0006 | < .0001 |
| Natural Direct Effect (NDE) | 0.0004 | 0.0001 | 0.0002 | 0.0006 | < .0001 |
| Natural Indirect Effect (NIE) | 0.0001 | 0.0000 | 0.0001 | 0.0001 | < .0001 |
| Percentage Mediated | 17.3526 | 3.7902 | 9.9241 | 24.7812 | < .0001 |
| Percentage Due to Interaction | 0 | . | . | . | . |
| Percentage Eliminated | 17.3526 | 3.7902 | 9.9241 | 24.7812 | < .0001 |

infectivity status from 0 to 20 years, after which increasing age is associated with a gradual increase in secondary infection. Table 3 summarizes the results from the CAUSALMED procedure. The proportion of the NDE was 82.65%, indicating that the NIE or the percentage mediated was 17.35%.

Clusters for those with severe dengue attributable to primary infection were primarily found in the northeastern region, while secondary infection clusters were mainly in the southeastern region (S6A and S6B Fig). Moran's I for those with severe dengue who had primary infection was 0.1129. With a critical z-score of 2.72 (p-value = 0.0065), there was less than 1% likelihood that the clustered pattern was due to chance. However, the pattern observed for the clusters who had secondary infection among those with severe dengue did not appear to be significantly different from random (Moran's I = -0.012, z-score = -0.5351, p-value = 0.5926). The Northeast region had more clusters for severe dengue among those with primary infection. Furthermore, bivariate analysis from S2 Table shows that, when compared to all regions, other than the Southeast, DENV-3 was most predominant in the northeast region.

## Discussion

There was a nonlinear downward trend from 0–10 years and an upward trend between those greater than 10 years of age and dengue severity. However, the effect of age on severe dengue was no longer significant for those 10 to 60 years after infectivity status was introduced into the causal model. The mediation analysis also revealed that infectivity status partially mediated the causal pathway between age and severe dengue by 17%. This suggests that assessing an individual's risk for secondary infection may be more helpful in dengue prevention initiatives. The steep decline of dengue fever severity from 0 to 10 years, and the steep increase for those 60 years and over, further suggests that infants and individuals 60 years and older will need close monitoring and possibly critical management of dengue infection. Two things common to the Northeastern region between 2012 and 2017 were the relatively higher prevalence of DENV-3 compared to other serotypes and the presence of severe dengue clusters among individuals with primary infection. While efforts must be taken to halt the secondary transmission of DF, rigorous measures in areas newer to DF spread must be considered to address DF complications. Surveillance among dengue-naïve populations should pay particular attention to the introduction of DENV-3 serotypes.

Children below 14 years and adults older than 50 years have been found to have a higher death and hospitalization rate [19]. A similar trend was seen in our study population among individuals with secondary infection, where the frequency of dengue severity slightly decreases in the early years of life and increases again from 60 years. The association between higher age and secondary infection is consistent with the literature [1]. Furthermore, the association between severe dengue and ageing populations [1] may be explained by the higher propensity of exposure to a second infection as a person ages. When an individual develops ADE from initial infection [34], cross-reaction with the second infecting virus results in immune complexes that result in cytokine release later in the infection process, further leading to vascular permeability, shock, and increasing the possibility of death [34].

Characterization of DENV-2 in the Southeastern region has been associated with the American/Asian genotypes from Southeast Asia [35]. Since the Southeastern region had the highest proportion of DENV-2, DENV-3, and DENV-4 serotypes, the association between the southeastern region and dengue severity clusters among those with secondary infection may be attributable to the prevalence of old circulating genotypes of the DENV-2, DENV-3, and DENV-4 serotypes in the region.

Apart from the Southeastern region, DENV-3 was also prevalent in the Northeastern region. A map showing the distribution of DENV serotypes across Mexico [36] has characterized hotspots of DENV-3 in both Northeastern and Southeastern Mexico. The association between the northeastern region and severe dengue among those with primary infection suggests a possible link between the predominantly circulating serotype and disease severity. DENV-3 has been associated with severe dengue for individuals with primary infection, while secondary infection with DENV-1, DENV-2, and DENV-4 are primarily associated with severe dengue [21]. Furthermore, the significant clustering of dengue severity among those with primary infection in the Northwestern region may be related to the migration of DENV serotypes towards the north as climate change may favor *Aedes aegypti* dissemination towards central and northern regions [37].

The association between increasing age and dengue severity among individuals with both primary and secondary infection has been established [1]. However, in our study, this was particularly true for individuals older than 60 years, and false for individuals between 0–10 years of age. Furthermore, after adding infectivity status into the model, the relationship between age and disease severity was no longer significant for individuals 10–60 years of age. This study shows that, rather than a linear relationship, extreme age groups are at an increased risk of severe dengue and this relationship is partly mediated by infectivity status. While age would indicate more years at risk and an increased likelihood of severe infection, it has been suggested that infants within 6–12 months may be at risk of severe dengue due to placental transfer of maternal IgG [38].

This study has several limitations. First, some studies have linked the history of two or more previous infections with a decreased risk of a new infection [8]. This hypothesis could not be tested in our population since information about several prior dengue infections in the past was unavailable. Secondly, there was a vast amount of missingness, particularly for the serotype data. However, the final sample size was 52,212, and 827 serotype records were available for analysis, with 9577 missing severe dengue (97.47%) classification and 41809 missing non-severe dengue (98.64%) classification (Table 2). Missingness was assumed to be random since the proportion of missingness across disease severity and covariates were proportional, and a complete case analysis was performed for all regression models. Lastly, the CAU-SALMED package is limited in its ability to perform sensitivity analysis to determine if the sequential ignorability assumption made in the causal analysis [39] is robust. Although the assumption has been considered non-refutable because it cannot be directly tested from observed data, our exposure and mediating variables are unlikely to compromise the assumption as much as parameters that may involve non-randomization. For instance, people do not assign themselves an 'age' nor choose whether to be classified as having a primary or secondary infection.

## Conclusion

Understanding the causal pathways and mechanisms with which age affects dengue severity allows targeting the right population group for interventions. The mediation effect of secondary infection status on dengue severity will enable scientists to focus on ways to prevent dengue re-infection and active management of patients with prior history of dengue, as well as in dengue-naïve populations. Primary infection in Northeastern Mexico and secondary infection in other regions of Mexico require continued surveillance. Furthermore, through the indirect effect of secondary infection, infants and the aged population are at an increased risk of dengue fever complications and requires critical care and management.

## Supporting information

**S1 Fig. Age distribution for different infection status categories.**
(TIF)

**S2 Fig.** a: Frequency polygon of Age pattern for severe dengue. b: Frequency polygon of Age pattern for non-severe dengue.
(TIF)

**S3 Fig.** a: Trend of infection status from 2012 to 2017 for severe dengue. b: Trend of infection status from 2012 to 2017 for non-severe dengue.
(TIF)

**S4 Fig. Plot of infection status by dengue severity for each region.**
(TIF)

**S5 Fig. Graph of serotype distribution by sex and dengue severity.**
(TIF)

**S6 Fig.** a: Severe Dengue Hotspot among individuals with primary infection. b: Severe Dengue Hotspot among individuals with secondary infection. Map: https://www.diva-gis.org/gdata.
(TIF)

**S1 Table. Demographic characteristics by infection classification.**
(DOCX)

**S2 Table. Relationship between the region and serotype distribution.**
(DOCX)

**S3 Table. Generalized additive model of the effect of age on infectivity status showing covariate results.**
(DOCX)

**S4 Table. Generalized additive model of the effect of age on dengue severity showing covariate results.**
(DOCX)

**S5 Table. Generalized additive model of the effect of age and infectivity status on dengue severity showing covariate results.**
(DOCX)

## Author Contributions

**Conceptualization:** Esther Annan, Bingxin Zhao.

**Data curation:** Esther Annan.

**Formal analysis:** Esther Annan.

**Methodology:** Esther Annan, Jesús Treviño, Bingxin Zhao, Alfonso J. Rodriguez-Morales, Ubydul Haque.

**Supervision:** Ubydul Haque.

**Visualization:** Esther Annan, Jesús Treviño, Ubydul Haque.

**Writing – original draft:** Esther Annan.

**Writing – review & editing:** Esther Annan, Jesús Treviño, Bingxin Zhao, Alfonso J. Rodriguez-Morales, Ubydul Haque.

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
