## [Decision Letter · Decision Letter 0]

31 Jan 2023

Dear Dr Annan,

Thank you very much for submitting your manuscript "Direct and indirect effects of age on dengue severity: the mediating role of secondary infection" for consideration at PLOS Neglected Tropical Diseases. As with all papers reviewed by the journal, your manuscript was reviewed by members of the editorial board and by several independent reviewers. In light of the reviews (below this email), we would like to invite the resubmission of a significantly-revised version that takes into account the reviewers' comments. 

We cannot make any decision about publication until we have seen the revised manuscript and your response to the reviewers' comments. Your revised manuscript is also likely to be sent to reviewers for further evaluation.

Sincerely,

Alberto Novaes Ramos Jr

Academic Editor

Andrea Marzi

Section Editor

Reviewer's Responses to Questions

**Key Review Criteria Required for Acceptance?**

**Methods**

-Are the objectives of the study clearly articulated with a clear testable hypothesis stated?

-Is the study design appropriate to address the stated objectives?

-Is the population clearly described and appropriate for the hypothesis being tested?

-Is the sample size sufficient to ensure adequate power to address the hypothesis being tested?

-Were correct statistical analysis used to support conclusions?

-Are there concerns about ethical or regulatory requirements being met?

Reviewer #1: 1. The sample size is sufficient to ensure adequate power to address the hypothesis test. However, some questions need to be clarified. The author need to check the equations in the mediation analysis. In line 169, the author wrote the total relative effect may also be calculated by summing up the relative direct and indirect effects (6) фi = θi + ᵦi b. However, the author calculated the total effect in equations (10-12) as Total effect = фi + ᵦi b. For example, - 0.1959 was ф1 not θ1 in equation 10. And the фi ≠ θi + ᵦi b in table 4. Maybe I am misunderstanding how these models work, but then I think it would be helpful to walk the reader through this a bit more.

2. There should also be a line between age and disease severity in Figure 1. Because according to existing information, secondary infection is not a complete mediator variable between age and severe dengue.

Reviewer #2: - The biggest question I have (which I think would help with the rest of the paper), is: why was a mediation analysis chosen to answer the research question of interest? Instead of finding the direct and indirect effects of age on dengue severity through secondary infection status, it sounds more like the research group is trying to see how age moderates the effect of secondary infection status on dengue severity. I would consider a moderation analysis, instead. If not, I would suggest making more explicit 1) the research question of interest, and hypothesized answers for that question; and 2) why a mediation analysis was chosen to answer this research question.

- I would suggest making more explicit 1) the population of interest, 2) how the sample was drawn from this population.

- The sample size was not mentioned in the Methods section (I would suggest doing that there); however, given the large N (approximately 52,000), I would think there is sufficient power for this analysis. I would suggest a sentence confirming this.

- Additionally, there was little mention of the Spatial Analysis (e.g., in the Introduction, it was not mentioned), and the rationale for including it was not clear.

Reviewer #3: The objective of the study is clearly stated with a testable hypothesis stated. The study design using a cross-sectional method is appropriate for the objectives. The population has been clearly defined and described with adequate sample sizes. The statistical analysis used to support conclusions are appropriately conducted. Since secondary data from public database was used there is no concern regarding ethical or regulatory requirements.

Reviewer #4: INTRODUCTION

Line 88/89 - I suggest using the current classification of dengue cases and not the old one in this part of the introduction section: “Of the people with a secondary infection, 2% - 3% progress to dengue hemorrhagic fever (DHF), which may further result in dengue shock syndrome (DSS) and death [17].”

METHODS

The description of spatial analysis is not well detailed. They used the Anselin Local Moran's I (LMi Index) statistic, but it is not clear how this method works. I suggest to the authors to describe this method more carefully.

**Results**

-Does the analysis presented match the analysis plan?

-Are the results clearly and completely presented?

-Are the figures (Tables, Images) of sufficient quality for clarity?

Reviewer #1: 1. What did Percent of Total Frequency mean in Fig S2&S3?

2. It is recommended to do statistical tests in Fig S2&S3, such as trend tests and comparison tests between groups.

3. Figure S7 was not found.

Reviewer #2: - The analysis presented matches the analysis plan; however, although the authors hoped to estimate direct and indirect effects, estimates of these effects were not explicitly presented.

- The figures were not numbered or labeled. I would suggest adding captions to all figures and tables.

- I would suggest adding a figure that illustrates the results.

Reviewer #3: Yes

Reviewer #4: The authors did not show the clusters results and just added a comment in the discussion section. They mentioned municipalities in the method, but presented analysis by regions (tables). I consider necessary to present a figure (map) to visualize the clusters. This will provide to the readers a better understanding about the distribution pattern in the study area. If possible, I also suggest presenting a map showing the distribution of cases by serotypes (DENV1, DENV2, DENV3 and DENV4).

**Conclusions**

-Are the conclusions supported by the data presented?

-Are the limitations of analysis clearly described?

-Do the authors discuss how these data can be helpful to advance our understanding of the topic under study?

-Is public health relevance addressed?

Reviewer #1: The authors should carefully check the results of the mediation analysis.

Reviewer #2: - There is a statement that the effect of age on dengue severity is primarily mediated by secondary infection status. I'm not sure the authors can say this unless they have the percentage of the total effect that is due to the indirect effect, and that percentage is substantially greater than 50%; I did not see that anywhere in the paper.

Reviewer #3: Yes

Reviewer #4: DISCUSSION

Line 289 – In this sentence, it is not clear if the analysis refers to clusters of severe cases of DENV-3 with primary infection, or if they only associate the higher prevalence of DENV-3 in the region with clusters of severe cases of primary infection.

**Editorial and Data Presentation Modifications?**

Reviewer #1: 1. According to existing studies, the elderly and children are high-risk groups for dengue fever. If the control group is set as the young people, the results may be more robust.

2. Much of the article described the region differences of severe dengue, but why did the author not adjust it in the model? 

3. Some factors, such as eduaction, income, diabetes, hypertension, that may affect the results should also be controlled for in the mediation analysis.

Reviewer #2: Please see PDF

Reviewer #3: None

Reviewer #4: (No Response)

**Summary and General Comments**

Reviewer #1: The authors have attempted to answer an important question about the severe dengue: if the effect of age on dengue severity was mediated by infectivity status. While the intention is nice, there are several issues that may influence whether the paper is worth publishing.

Reviewer #2: I think this article has potential to contribute to the dengue literature and public health research at large; however, I think much work needs to be done on defining the research question, being explicit about the population and sample of interest, choosing an analysis (or analyses) that answers that research question, and presenting results that correspond to the chosen analysis.

Reviewer #3: None

Reviewer #4: The study addresses a significant issue for the understanding of factors associated with dengue severity, especially age and secondary infection. The methods are adequate to answer the study questions and it was well performed.

PLOS authors have the option to publish the peer review history of their article (what does this mean?). If published, this will include your full peer review and any attached files.

Reviewer #1: No

Reviewer #2: No

Reviewer #3: Yes: Denny John

Reviewer #4: No
---

## [Decision Letter · Decision Letter 1]

18 Apr 2023

Dear Dr Annan,

Thank you very much for submitting your manuscript "Direct and indirect effects of age on dengue severity: the mediating role of secondary infection" for consideration at PLOS Neglected Tropical Diseases. As with all papers reviewed by the journal, your manuscript was reviewed by members of the editorial board and by several independent reviewers. In light of the reviews (below this email), we would like to invite the resubmission of a significantly-revised version that takes into account the reviewers' comments. 

We cannot make any decision about publication until we have seen the revised manuscript and your response to the reviewers' comments. Your revised manuscript is also likely to be sent to reviewers for further evaluation.

Sincerely,

Alberto Novaes Ramos Jr

Academic Editor

Andrea Marzi

Section Editor

Reviewer's Responses to Questions

**Key Review Criteria Required for Acceptance?**

**Methods**

-Are the objectives of the study clearly articulated with a clear testable hypothesis stated?

-Is the study design appropriate to address the stated objectives?

-Is the population clearly described and appropriate for the hypothesis being tested?

-Is the sample size sufficient to ensure adequate power to address the hypothesis being tested?

-Were correct statistical analysis used to support conclusions?

-Are there concerns about ethical or regulatory requirements being met?

Reviewer #1: (No Response)

Reviewer #2: Please see track-changed document.

Reviewer #3: Methods are clear and presented in a scientific manner

Reviewer #4: The authors provided more detail on the spatial analysis in the methods section, as suggested.

**Results**

-Does the analysis presented match the analysis plan?

-Are the results clearly and completely presented?

-Are the figures (Tables, Images) of sufficient quality for clarity?

Reviewer #1: (No Response)

Reviewer #2: Please see track-changed document.

Reviewer #3: Yes. the results presented match the analysis plan.

Reviewer #4: The authors explained and described the results of the cluster analyzes as suggested.

**Conclusions**

-Are the conclusions supported by the data presented?

-Are the limitations of analysis clearly described?

-Do the authors discuss how these data can be helpful to advance our understanding of the topic under study?

-Is public health relevance addressed?

Reviewer #1: (No Response)

Reviewer #2: Please see track-changed document.

Reviewer #3: The conclusions support the data presented. Limitations are clearly described. Public relevance is addressed.

Reviewer #4: The authors adjusted the text to be more explicit, as suggested.

**Editorial and Data Presentation Modifications?**

Reviewer #1: (No Response)

Reviewer #2: (No Response)

Reviewer #3: Accept

Reviewer #4: (No Response)

**Summary and General Comments**

Reviewer #1: 1. The subject of this research is very interesting. However, I am skeptical about the results due to contradictory results in the analysis. For example, Table 4 reports a statistically significant negative total effect, and the total effect calculated manually using Equations 12-14 should have been consistent with the result generated by SAS in Table 4. However, this paper presents inconsistent or even completely opposite results.

2. The primary aim of this paper is to examine the mediating effect of secondary infection history on the association between age and dengue severity. However, the relevant results are limited, and most of the findings presented are not directly related to the research question, such as those in the Spatial Analysis section.

3. In this paper, infants under one year old are utilized as the control group. However, the number of individuals in other age groups is significantly higher than that of this control group, exceeding it by more than tenfold. This may seriously impact the statistical power of the analysis.

Reviewer #2: Please see track-changed document.

Reviewer #3: None

Reviewer #4: (No Response)

PLOS authors have the option to publish the peer review history of their article (what does this mean?). If published, this will include your full peer review and any attached files.

Reviewer #1: No

Reviewer #2: No

Reviewer #3: Yes: Denny John

Reviewer #4: No
---

## [Decision Letter · Decision Letter 2]

14 Jun 2023

Dear Dr Annan,

Thank you very much for submitting your manuscript "Direct and indirect effects of age on dengue severity: the mediating role of secondary infection" for consideration at PLOS Neglected Tropical Diseases. As with all papers reviewed by the journal, your manuscript was reviewed by members of the editorial board and by several independent reviewers. The reviewers appreciated the attention to an important topic. Based on the reviews, we are likely to accept this manuscript for publication, providing that you modify the manuscript according to the review recommendations. 

Sincerely,

Alberto Novaes Ramos Jr

Academic Editor

Andrea Marzi

Section Editor

Reviewer's Responses to Questions

**Key Review Criteria Required for Acceptance?**

**Methods**

-Are the objectives of the study clearly articulated with a clear testable hypothesis stated?

-Is the study design appropriate to address the stated objectives?

-Is the population clearly described and appropriate for the hypothesis being tested?

-Is the sample size sufficient to ensure adequate power to address the hypothesis being tested?

-Were correct statistical analysis used to support conclusions?

-Are there concerns about ethical or regulatory requirements being met?

Reviewer #1: (No Response)

Reviewer #2: (No Response)

**Results**

-Does the analysis presented match the analysis plan?

-Are the results clearly and completely presented?

-Are the figures (Tables, Images) of sufficient quality for clarity?

Reviewer #1: (No Response)

Reviewer #2: (No Response)

**Conclusions**

-Are the conclusions supported by the data presented?

-Are the limitations of analysis clearly described?

-Do the authors discuss how these data can be helpful to advance our understanding of the topic under study?

-Is public health relevance addressed?

Reviewer #1: (No Response)

Reviewer #2: (No Response)

**Editorial and Data Presentation Modifications?**

Reviewer #1: (No Response)

Reviewer #2: (No Response)

**Summary and General Comments**

Reviewer #1: It is advisable to consult with epidemiologists to review the analysis and the results of this study.

Reviewer #2: Per a previous comment, I was still not able to see the figures, but assuming those are correct, all comments were addressed.

PLOS authors have the option to publish the peer review history of their article (what does this mean?). If published, this will include your full peer review and any attached files.

Reviewer #1: No

Reviewer #2: No

Figure Files:

Data Requirements:

Reproducibility:

References

---

## [Decision Letter · Decision Letter 3]

17 Jul 2023

Dear Dr Annan,

We are pleased to inform you that your manuscript 'Direct and indirect effects of age on dengue severity: the mediating role of secondary infection' has been provisionally accepted for publication in PLOS Neglected Tropical Diseases.

Best regards,

Alberto Novaes Ramos Jr

Academic Editor

Andrea Marzi

Section Editor

Reviewer's Responses to Questions

**Key Review Criteria Required for Acceptance?**

**Methods**

-Are the objectives of the study clearly articulated with a clear testable hypothesis stated?

-Is the study design appropriate to address the stated objectives?

-Is the population clearly described and appropriate for the hypothesis being tested?

-Is the sample size sufficient to ensure adequate power to address the hypothesis being tested?

-Were correct statistical analysis used to support conclusions?

-Are there concerns about ethical or regulatory requirements being met?

Reviewer #1: (No Response)

**Results**

-Does the analysis presented match the analysis plan?

-Are the results clearly and completely presented?

-Are the figures (Tables, Images) of sufficient quality for clarity?

Reviewer #1: (No Response)

**Conclusions**

-Are the conclusions supported by the data presented?

-Are the limitations of analysis clearly described?

-Do the authors discuss how these data can be helpful to advance our understanding of the topic under study?

-Is public health relevance addressed?

Reviewer #1: (No Response)

**Editorial and Data Presentation Modifications?**

Reviewer #1: (No Response)

**Summary and General Comments**

Reviewer #1: (No Response)

PLOS authors have the option to publish the peer review history of their article (what does this mean?). If published, this will include your full peer review and any attached files.

Reviewer #1: No

---

## [Editor Report · Acceptance letter]

3 Aug 2023

Dear Dr Annan,

We are delighted to inform you that your manuscript, "Direct and indirect effects of age on dengue severity: the mediating role of secondary infection," has been formally accepted for publication in PLOS Neglected Tropical Diseases.

Best regards,

Shaden Kamhawi

co-Editor-in-Chief

Paul Brindley

co-Editor-in-Chief
